# Linear-Memory and Decomposition-Invariant Linearly Convergent Conditional Gradient Algorithm for Structured Polytopes

**Dan Garber**
Toyota Technological Institute at Chicago
dgarber@ttic.edu

**Ofer Meshi**
Google
meshi@google.com

## Abstract

Recently, several works have shown that natural modifications of the classical conditional gradient method (aka Frank-Wolfe algorithm) for constrained convex optimization, provably converge with a linear rate when: i) the feasible set is a polytope, and ii) the objective is smooth and strongly-convex. However, all of these results suffer from two significant shortcomings:

1. large memory requirement due to the need to store an explicit convex decomposition of the current iterate, and as a consequence, large running-time overhead per iteration
2. the worst case convergence rate depends unfavorably on the dimension

In this work we present a new conditional gradient variant and a corresponding analysis that improves on both of the above shortcomings. In particular:

1. both memory and computation overheads are only linear in the dimension
2. in case the optimal solution is sparse, the new convergence rate replaces a factor which is at least linear in the dimension in previous work, with a linear dependence on the number of non-zeros in the optimal solution

At the heart of our method and corresponding analysis, is a novel way to compute decomposition-invariant *away-steps*. While our theoretical guarantees do not apply to any polytope, they apply to several important structured polytopes that capture central concepts such as paths in graphs, perfect matchings in bipartite graphs, marginal distributions that arise in structured prediction tasks, and more. Our theoretical findings are complemented by empirical evidence which shows that our method delivers state-of-the-art performance.

## 1 Introduction

The efficient reduction of a constrained convex optimization problem to a constrained linear optimization problem is an appealing algorithmic concept, in particular for large-scale problems. The reason is that for many feasible sets of interest, the problem of minimizing a linear function over the set admits much more efficient methods than its non-linear convex counterpart. Prime examples for this phenomenon include various structured polytopes that arise in combinatorial optimization, such as the path polytope of a graph (aka the unit flow polytope), the perfect matching polytope of a bipartite graph, and the base polyhedron of a matroid, for which we have highly efficient combinatorial algorithms for linear minimization that rely heavily on the specific rich structure of the polytope [21]. At the same time, minimizing a non-linear convex function over these sets usually requires the use of generic interior point solvers that are oblivious to the specific combinatorial structure of the underlying set, and as a result, are often much less efficient. Indeed, it is for this reason, that the

conditional gradient (CG) method (aka Frank-Wolfe algorithm), a method for constrained convex optimization that is based on solving linear subproblems over the feasible domain, has regained much interest in recent years in the machine learning, signal processing and optimization communities. It has been recently shown that the method delivers state-of-the-art performance on many problems of interest, see for instance [14, 17, 4, 10, 11, 22, 19, 25, 12, 15].

As part of the regained interest in the conditional gradient method, there is also a recent effort to understand the convergence rates and associated complexities of conditional gradient-based methods, which are in general far less understood than other first-order methods, e.g., the projected gradient method. It is known, already from the first introduction of the method by Frank and Wolfe in the 1950's [5] that the method converges with a rate of roughly $O(1/t)$ for minimizing a smooth convex function over a convex and compact set. However, it is not clear if this convergence rate improves under an additional standard strong-convexity assumption. In fact, certain lower bounds, such as in [18, 8], suggest that such improvement, even if possible, should come with a worse dependence on the problem's parameters (e.g., the dimension). Nevertheless, over the past years, various works tried to design natural variants of the CG method that converge provably faster under the strong convexity assumption, without dramatically increasing the per-iteration complexity of the method. For instance, GuéLat and Marcotte [9] showed that a CG variant which uses the concept of *away-steps* converges exponentially fast in case the objective function is strongly convex, the feasible set is a polytope, and the optimal solution is located in the interior of the set. A similar result was presented by Beck and Teboulle [3] who considered a specific problem they refer to as *the convex feasibility problem* over an arbitrary convex set. They also obtained a linear convergence rate under the assumption that an optimal solution that is far enough from the boundary of the set exists. In both of these works, the exponent depends on the distance of the optimal solution from the boundary of the set, which in general can be arbitrarily small. Later, Ahipasaoglu et al. [1] showed that in the specific case of minimizing a smooth and strongly convex function over the unit simplex, a variant of the CG method which also uses away-steps, converges with a linear rate. Unfortunately, it is not clear from their analysis how this rate depends on natural parameters of the problem such as the dimension and the condition number of the objective function.

Recently, Garber and Hazan presented a linearly-converging CG variant for polytopes without any restrictions on the location of the optimum [8]. In a later work, Lacoste-Julien and Jaggi [16] gave a refined affine-invariant analysis of an algorithm presented in [9] which also uses away steps, and showed that it also converges exponentially fast in the same setting as the Garber-Hazan result. More recently, Beck and Shtern [2] gave a different, duality-based, analysis for the algorithm of [9], and showed that it can be applied to a wider class of functions than purely strongly convex functions. However, the explicit dependency of their convergence rate on the dimension is suboptimal, compared to [8, 16]. Aside from the polytope case, Garber and Hazan [7] have shown that in case the feasible set is strongly-convex and the objective function satisfies certain strong convexity-like proprieties, then the standard CG method converges with an accelerated rate of $O(1/t^2)$. Finally, in [6] Garber showed a similar improvement (roughly quadratic) for the *spectrahedron* – the set of unit trace positive semidefinite matrices.

Despite the exponential improvement in convergence rate for polytopes obtained in recent results, they all suffer from two major drawbacks. First, while in terms of the number of calls per-iteration to the linear optimization oracle, these methods match the standard CG method, i.e., a single call per iteration, the overhead of other operations both in terms of running time and memory requirements is significantly worse. The reason is that in order to apply the so-called away-steps, which all methods use, they require to maintain at all times an explicit decomposition of the current iterate into vertices of the polytope. In the worst case, maintaining such a decomposition and computing the away-steps require both memory and per-iteration runtime overheads that are at least quadratic in the dimension. This is much worse than the standard CG method, whose memory and runtime overheads are only linear in the dimension. Second, the convergence rate of all previous linearly convergent CG methods depends explicitly on the dimension. While it is known that this dependency is unavoidable in certain cases, e.g., when the optimal solution is, informally speaking, dense (see for instance the lower bound in [8]), it is not clear that such an unfavorable dependence is mandatory when the optimum is sparse.

In this paper, we revisit the application of CG variants to smooth and strongly-convex optimization over polytopes. We introduce a new variant which overcomes both of the above shortcomings from which all previous linearly-converging variants suffer. The main novelty of our method, which is the key to its improved performance, is that unlike previous variants, it is decomposition-invariant, i.e., it

| Paper | #iterations to $\epsilon$ err. | #LOO calls | runtime | memory |
|---|---|---|---|---|
| Frank & Wolfe [5] | $\beta D^2/\epsilon$ | 1 | $n$ | $n$ |
| Garber & Hazan [8] | $\kappa n D^2 \log(1/\epsilon)$ | 1 | $n\min(n,t)$ | $n\min(n,t)$ |
| Lacoste-Julien & Jaggi [16] | $\kappa n D^2 \log(1/\epsilon)$ | 1 | $n\min(n,t)$ | $n\min(n,t)$ |
| Beck & Shtern [2] | $\kappa n^2 D^2 \log(1/\epsilon)$ | 1 | $n\min(n,t)$ | $n\min(n,t)$ |
| This paper | $\kappa \mathrm{card}(x^*)D^2 \log(1/\epsilon)$ | 2 | $n$ | $n$ |

Table 1: Comparison with previous work. We define $\kappa := \beta/\alpha$, we let $n$ denote the dimension and $D$ denote the Euclidean diameter of the polytope. The third column gives the number of calls to the linear optimization oracle per iteration, the fourth column gives the additional arithmetic complexity at iteration $t$, and the fifth column gives the worst case memory requirement at iteration $t$. The bounds for the algorithms of [8, 16, 2], which are independent of $t$, assume an algorithmic version of Carathéodory's theorem, as fully detailed in [2]. The bound on number of iterations of [16] depends on the squared inverse *pyramidal width* of $\mathcal{P}$, which is difficult to evaluate, however, this quantity is at least proportional to $n$.

does not require to maintain an explicit convex decomposition of the current iterate. This principle proves to be crucial both for eliminating the memory and runtime overheads, as well as to obtaining shaper convergence rates for instances that admit a sparse optimal solution.

A detailed comparison of our method to previous art is shown in Table 1. We also provide in Section 5 empirical evidence that the proposed method delivers state-of-the-art performance on several tasks of interest. While our method is less general than previous ones, i.e., our theoretical guarantees do not hold for arbitrary polytopes, they readily apply to many structured polytopes that capture important concepts such as paths in graphs, perfect matchings in bipartite graphs, Markov random fields, and more.

## 2 Preliminaries

Throughout this work we let $\|\cdot\|$ denote the standard Euclidean norm. Given a point $x \in \mathbb{R}^n$, we let $\mathrm{card}(x)$ denote the number of non-zero entries in $x$.

**Definition 1.** *We say that a function $f(x) : \mathbb{R}^n \to \mathbb{R}$ is $\alpha$-strongly convex w.r.t. a norm $\|\cdot\|$, if for all $x,y \in \mathbb{R}^n$ it holds that $f(y) \geq f(x) + \nabla f(x) \cdot (y-x) + \frac{\alpha}{2}\|x-y\|^2$.*

**Definition 2.** *We say that a function $f(x) : \mathbb{R}^n \to \mathbb{R}$ is $\beta$-smooth w.r.t. a norm $\|\cdot\|$, if for all $x,y \in \mathbb{R}^n$ it holds that $f(y) \leq f(x) + \nabla f(x) \cdot (y-x) + \frac{\beta}{2}\|x-y\|^2$.*

The first-order optimality condition implies that for a $\alpha$-strongly convex $f$, if $x^*$ is the unique minimizer of $f$ over a convex and compact set $\mathcal{K} \subset \mathbb{R}^n$, then for all $x \in \mathcal{K}$ it holds that

$$f(x) - f(x^*) \geq \frac{\alpha}{2}\|x - x^*\|^2. \tag{1}$$

### 2.1 Setting

In this work we consider the optimization problem $\min_{x \in \mathcal{P}} f(x)$, where we assume that:

- $f(x)$ is $\alpha$-strongly convex and $\beta$-smooth with respect to the Euclidean norm.
- $\mathcal{P}$ is a polytope which satisfies the following two properties:
    1. $\mathcal{P}$ can be described algebraically as $\mathcal{P} = \{x \in \mathbb{R}^n \,|\, x \geq 0, \ Ax = b\}$.
    2. All vertices of $\mathcal{P}$ lie on the hypercube $\{0,1\}^n$.

We let $x^*$ denote the (unique) minimizer of $f$ over $\mathcal{P}$, and we let $D$ denote the Euclidean diameter of $\mathcal{P}$, namely, $D = \max_{x,y \in \mathcal{P}} \|x - y\|$. We let $\mathcal{V}$ denote the set of vertices of $\mathcal{P}$, where according to our assumptions, it holds that $\mathcal{V} \subseteq \{0,1\}^n$.

While the polytopes that satisfy the above assumptions are not completely general, these assumptions already capture several important concepts such as paths in graphs, perfect-matchings, Markov

random fields, and more. Indeed, a surprisingly large number of applications from machine learning, signal processing and other domains are formulated as optimization problems in this category (e.g., [13, 15, 16]). We give detailed examples of such polytopes in Section A in the appendix. Importantly, the above assumptions allow us to get rid of the dependency of the convergence rate on certain geometric parameters (such as $\psi, \xi$ in [8] or the *pyramidal width* in [16]), which can be polynomial in the dimension, and hence result in an impractical convergence rate. Finally, for many of these polytopes, the vertices are sparse, i.e., for any vertex $v \in \mathcal{V}$, $\text{card}(v) << n$. In this case, when the optimum $x^*$ can be decomposed as a convex combination of only a few vertices (and thus, sparse by itself), we get a sharper convergence rate that depends on the sparsity of $x^*$ and not explicitly on the dimension, as in previous work. We believe that our theoretical guarantees could be well extended to more general polytopes, as suggested in Section C in the appendix; we leave this extension for future work.

## 3 Our Approach

In order to better communicate our ideas, we begin by first briefly introducing the standard conditional gradient method and its accelerated away-steps-based variants. We discuss both the blessings and shortcomings of these away-steps-based variants in Subsection 3.1. Then, in Subsection 3.2, we present our new method, a decomposition-invariant away-steps-based conditional gradient algorithm, and discuss how it addresses the shortcomings of previous variants.

### 3.1 The conditional gradient method and acceleration via away-steps

The standard conditional gradient algorithm is given below (Algorithm 1). It is well known that when setting the step-size $\eta_t$ in an appropriate way, the worst case convergence rate of the method is $O(\beta D^2/t)$ [13]. This convergence rate is tight for the method in general, see for instance [18].

---

**Algorithm 1** Conditional Gradient

1: let $x_1$ be some vertex in $\mathcal{V}$
2: **for** $t = 1...$ **do**
3:      $v_t \leftarrow \arg\min_{v \in \mathcal{V}} v \cdot \nabla f(x_t)$
4:      choose a step-size $\eta_t \in (0, 1]$
5:      $x_{t+1} \leftarrow x_t + \eta_t(v_t - x_t)$
6: **end for**

---

**Algorithm 2** Pairwise Conditional Gradient

1: let $x_1$ be some vertex in $\mathcal{V}$
2: **for** $t = 1...$ **do**
3:      let $\sum_{i=1}^{k_t} \lambda_t^{(i)} v_t^{(i)}$ be an **explicitly maintained** convex decomposition of $x_t$
4:      $v_t^+ \leftarrow \arg\min_{v \in \mathcal{V}} v \cdot \nabla f(x_t)$
5:      $j_t \leftarrow \arg\min_{j \in [k_t]} v_t^{(j)} \cdot (-\nabla f(x_t))$
6:      choose a step-size $\eta_t \in (0, \lambda_t^{(j_t)}]$
7:      $x_{t+1} \leftarrow x_t + \eta_t(v_t^+ - v_t^{(j_t)})$
8:      update the convex decomposition of $x_{t+1}$
9: **end for**

---

Consider the iterate of Algorithm 1 on iteration $t$, and let $x_t = \sum_{i=1}^{k} \lambda_i v_i$ be its convex decomposition into vertices of the polytope $\mathcal{P}$. Note that Algorithm 1 implicitly discounts each coefficient $\lambda_i$ by a factor $(1 - \eta_t)$, in favor of the new added vertex $v_t$. A different approach is not to decrease all vertices in the decomposition of $x_t$ uniformly, but to more-aggressively decrease vertices that are worse than others with respect to some measure, such as their product with the gradient direction. This key principle proves to be crucial to breaking the $1/t$ rate of the standard method, and to achieve a linear convergence rate under certain strong-convexity assumptions, as described in the recent works [8, 16, 2]. For instance, in [8] it has been shown, via the introduction of the concept of a *Local Linear Optimization Oracle*, that using such a non-uniform reweighing rule, in fact approximates a certain *proximal* problem, that together with the shrinking effect of strong convexity, as captured by Eq. (1), yields a linear convergence rate. We refer to these methods as away-step-based CG methods. As a concrete example, which will also serve as a basis for our new method, we describe the *pairwise* variant recently studied in [16], which applies this principle in Algorithm 2.[1] Note that Algorithm 2 decreases the weight of exactly one vertex in the decomposition: that with the largest product with the gradient.

It is important to note that since previous away-step-based CG variants do not decrease the coefficients in the convex decomposition of the current iterate uniformly, they all require to explicitly store and maintain a convex decomposition of the current iterate. This issue raises two main disadvantages:

**Superlinear memory and running-time overheads** Storing a decomposition of the current iterate as a convex combination of vertices generally requires $O(n^2)$ memory in the worst case. While the away-step-based variants increase the size of the decomposition by at most a single vertex per iteration, they also typically exhibit linear convergence after performing at least $\Omega(n)$ steps [8, 16, 2], and thus, this $O(n^2)$ estimate still holds. Moreover, since these methods require i) to find the worst vertex in the decomposition, in terms of dot-product with current gradient direction, and ii) to update this decomposition at each iteration (even when using sophisticated update techniques such as in [2]), then the worst case per-iteration overhead in terms of computation is also $\Omega(n^2)$.

**Decomposition-specific performance** The choice of away-step depends on the specific decomposition that is maintained by the algorithm. Since the feasible point $x_t$ may admit several different convex decompositions, committing to one such decomposition, might result in sub-optimal away-steps. As observable in Table 1, for certain problems in which the optimal solution is sparse, all analyses of previous away-steps-based variants are significantly suboptimal, since they all depend explicitly on the dimension. This seems to be an unavoidable side-effect of being decomposition-dependent. On the other hand, the fact that our new approach is decomposition-invariant allows us to obtain sharper convergence rates for such instances.

### 3.2 A new decomposition-invariant pairwise conditional gradient method

Our main observation is that in many cases of interest, given a feasible iterate $x_t$, one can in fact compute an optimal away-step from $x_t$ without relying on any single specific decomposition. This observation allows us to overcome both of the main disadvantages of previous away-step-based CG variants. Our algorithm, which we refer to as a *decomposition-invariant pairwise conditional gradient* (DICG), is given below in Algorithm 3.

---

**Algorithm 3** Decomposition-invariant Pairwise Conditional Gradient

1: input: sequence of step-sizes $\{\eta_t\}_{t \geq 1}$
2: let $x_0$ be an arbitrary point in $\mathcal{P}$
3: $x_1 \leftarrow \arg\min_{v \in \mathcal{V}} v \cdot \nabla f(x_0)$
4: **for** $t = 1...$ **do**
5: $\quad v_t^+ \leftarrow \arg\min_{v \in \mathcal{V}} v \cdot \nabla f(x_t)$
6: $\quad$ define the vector $\tilde{\nabla} f(x_t) \in \mathbb{R}^n$ as follows: $[\tilde{\nabla} f(x_t)]_i := \begin{cases} [\nabla f(x_t)]_i & \text{if } x_t(i) > 0 \\ -\infty & \text{if } x_t(i) = 0 \end{cases}$
7: $\quad v_t^- \leftarrow \arg\min_{v \in \mathcal{V}} v \cdot (-\tilde{\nabla} f(x_t))$
8: $\quad$ choose a new step-size $\tilde{\eta}_t$ using one of the following two options:
$\quad\quad$ **Option 1: predefined step-size**
$\quad\quad\quad\quad$ let $\delta_t$ be the smallest natural number such that $2^{-\delta_t} \leq \eta_t$, and set a new step-size $\tilde{\eta}_t \leftarrow 2^{-\delta_t}$
$\quad\quad$ **Option 2: line-search**
$\quad\quad\quad\quad \gamma_t \leftarrow \max_{\gamma \in [0,1]}\{x_t + \gamma(v_t^+ - v_t^-) \geq 0\}, \quad \tilde{\eta}_t \leftarrow \min_{\eta \in (0, \gamma_t]} f(x_t + \eta(v_t^+ - v_t^-))$
9: $\quad x_{t+1} \leftarrow x_t + \tilde{\eta}_t(v_t^+ - v_t^-)$
10: **end for**

---

The following observation shows the optimality of away-steps taken by Algorithm 3.

**Observation 1** (optimal away-steps in Algorithm 3). *Consider an iteration $t$ of Algorithm 3 and suppose that the iterate $x_t$ is feasible. Let $x_t = \sum_{i=1}^{k} \lambda_i v_i$ for some integer $k$, be an irreducible way of writing $x_t$ as a convex sum of vertices of $\mathcal{P}$, i.e., $\lambda_i > 0$ for all $i \in [k]$. Then it holds that $\forall i \in [k] : v_i \cdot \nabla f(x_t) \leq v_t^- \cdot \nabla f(x_t)$, and $\gamma_t \geq \min\{x_t(i) \mid i \in [n], x_t(i) > 0\}$.*

*Proof.* Let $x_t = \sum_{i=1}^{k} \lambda_i v_i$ be a convex decomposition of $x_t$ into vertices of $\mathcal{P}$, for some integer $k$, where each $\lambda_i$ is positive. Note that it must hold that for any $j \in [n]$ and any $i \in [k]$, $x_t(j) = 0 \Rightarrow v_i(j) = 0$, since by our assumption $\mathcal{V} \subset \mathbb{R}_+^n$. The observation then follows directly from the definition of $v_t^-$. $\square$

We next state the main theorem of this paper, which bounds the convergence rate of Algorithm 3. The proof is provided in Section B.3 in the appendix.

**Theorem 1.** *Let $M_1 = \sqrt{\alpha/(8\,card(x^*))}$ and $M_2 = \beta D^2/2$. Consider running Algorithm 3 with Option 1 as the step-size, and suppose that $\forall t \geq 1 : \eta_t = \left(M_1/(2\sqrt{M_2})\right)\left(1 - M_1^2/(4M_2)\right)^{\frac{t-1}{2}}$. Then, the iterates of Algorithm 3 are always feasible, and $\forall t \geq 1$:*

$$f(x_t) - f(x^*) \leq \frac{\beta D^2}{2}\exp\left(-\frac{\alpha}{8\beta D^2\,card(x^*)}t\right) \ .$$

We now turn to make several remarks regarding Algorithm 3 and Theorem 1:

The so-called *dual gap*, defined as $g_t := (x_t - v_t^+)\cdot\nabla f(x_t)$, which serves as a certificate for the sub-optimality of the iterates of Algorithm 3, also converges with a linear rate, as we prove in Section B.4 in the appendix.

Note that despite the different parameters of the problem at hand (e.g., $\alpha, \beta, D, card(x^*)$), running the algorithm with Option 1 for choosing the step-size, for which the guarantee of Theorem 1 holds, requires knowing a *single* parameter, i.e., $M_1/\sqrt{M_2}$. In particular, it is an easy consequence that running the algorithm with an estimate $M \in [0.5M_1/\sqrt{M_2}, M_1/\sqrt{M_2}]$, will only affect the leading constant in the convergence rate listed in the theorem. Hence, $M_1/\sqrt{M_2}$ could be efficiently estimated via a logarithmic-scale search.

Theorem 1 improves significantly over the convergence rate established for the pairwise conditional gradient variant in [16]. In particular, the number of iterations to reach an $\epsilon$ error in the analysis of [16] depends linearly on $|\mathcal{V}|!$, where $|\mathcal{V}|$ is the number of vertices of $\mathcal{P}$.

# 4 Analysis

Throughout this section we let $h_t$ denote the approximation error of Algorithm 3 on iteration $t$, for any $t \geq 1$, i.e., $h_t = f(x_t) - f(x^*)$.

## 4.1 Feasibility of the iterates generated by Algorithm 3

We start by proving that the iterates of Algorithm 3 are always feasible. While feasibility is straightforward when using the the line-search option to set the step-size (Option 2), it is less obvious when using Option 1. We will make use of the following observation, which is a simple consequence of the optimal choice of $v_t^-$ and our assumptions on $\mathcal{P}$. A proof is given in Section B.1 in the appendix.

**Observation 2.** *Suppose that on some iteration $t$ of Algorithm 3, the iterate $x_t$ is feasible, and that the step-size is chosen using Option 1. Then, if for all $i \in [n]$ for which $x_t(i) \neq 0$ it holds that $x_t(i) \geq \tilde{\eta}_t$, the following iterate $x_{t+1}$ is also feasible.*

**Lemma 1** (feasibility of iterates under Option 1)**.** *Suppose that the sequence of step-sizes $\{\eta_t\}_{t\geq 1}$ is monotonically non-increasing, and contained in the interval $[0, 1]$. Then, the iterates generated by Algorithm 3 using Option 1 for setting the step-size, are always feasible.*

*Proof.* We are going to prove by induction that on each iteration $t$ there exists a non-negative integer-valued vector $s_t \in \mathbb{N}^n$, such that for any $i \in [n]$, it holds that $x_t(i) = 2^{-\delta_t}s_t(i)$. The lemma then follows since, by definition, $\tilde{\eta}_t = 2^{-\delta_t}$, and by applying Observation 2. The base case $t = 1$ holds since $x_1$ is a vertex of $\mathcal{P}$ and thus for any $i \in [n]$ we have that $x_1(i) \in \{0, 1\}$ (recall that $\mathcal{V} \subset \{0, 1\}^n$). On the other hand, since $\eta_1 \leq 1$, it follows that $\delta_1 \geq 0$. Thus, there indeed exists a non-negative integer-valued vector $s_1$, such that $x_1 = 2^{-\delta_1}s_1$.

Suppose now that the induction holds for some $t \geq 1$. Since by definition of $v_t^-$, subtracting $\tilde{\eta}_t v_t^-$ from $x_t$ can only decrease positive entries in $x_t$ (see proof of Observation 2), and both $v_t^-, v_t^+$ are vertices of $\mathcal{P}$ (and thus in $\{0, 1\}^n$), and $\tilde{\eta}_t = 2^{-\delta_t}$, it follows that each entry $i$ in $x_{t+1}$ is given by:

$$x_{t+1}(i) = 2^{-\delta_t}\begin{cases} s_t(i) & \text{if } s_t(i) \geq 1 \ \& \ v_t^-(i) = v_t^+(i) = 1 \text{ or } v_t^-(i) = v_t^+(i) = 0 \\ s_t(i) - 1 & \text{if } s_t(i) \geq 1 \ \& \ v_t^-(i) = 1 \ \& \ v_t^+(i) = 0 \\ s_t(i) + 1 & \text{if } v_t^-(i) = 0 \ \& \ v_t^+(i) = 1 \end{cases}$$

Thus, $x_{t+1}$ can also be written in the form $2^{-\delta_t}\tilde{s}_{t+1}$ for some $\tilde{s}_{t+1} \in \mathbb{N}^n$. By definition of $\delta_t$ and the monotonicity of $\{\eta_t\}_{t\geq 1}$, we have that $\frac{2^{-\delta_t}}{2^{-\delta_{t+1}}}$ is a positive integer. Thus, setting $s_{t+1} = \frac{2^{-\delta_t}}{2^{-\delta_{t+1}}}\tilde{s}_{t+1}$, the induction holds also for $t + 1$. $\qquad\square$

## 4.2 Bounding the per-iteration error-reduction of Algorithm 3

The following technical lemma is the key to deriving the linear convergence rate of our method, and in particular, to deriving the improved dependence on the sparsity of $x^*$, instead of the dimension. At a high-level, the lemma translates the $\ell_2$ distance between two feasible points into a $\ell_1$ distance in a simplex defined over the set of vertices of the polytope.

**Lemma 2.** *Let $x, y \in \mathcal{P}$. There exists a way to write $x$ as a convex combination of vertices of $\mathcal{P}$, $x = \sum_{i=1}^{k} \lambda_i v_i$ for some integer $k$, such that $y$ can be written as $y = \sum_{i=1}^{k}(\lambda_i - \Delta_i)v_i + (\sum_{i=1}^{k} \Delta_i)z$ with $\Delta_i \in [0, \lambda_i] \, \forall i \in [k], z \in \mathcal{P}$, and $\sum_{i=1}^{k} \Delta_i \leq \sqrt{card(y)}\|x - y\|$.*

The proof is given in Section B.2 in the appendix. The next lemma bounds the per-iteration improvement of Algorithm 3 and is the key step to proving Theorem 1. We defer the rest of the proof of Theorem 1 to Section B.3 in the appendix.

**Lemma 3.** *Consider the iterates of Algorithm 3, when the step-sizes are chosen using Option 1. Let $M_1 = \sqrt{\alpha/(8card(x^*))}$ and $M_2 = \beta D^2/2$. For any $t \geq 1$ it holds that $h_{t+1} \leq h_t - \eta_t M_1 h_t^{1/2} + \eta_t^2 M_2$.*

*Proof.* Define $\Delta_t = \sqrt{2card(x^*)h_t/\alpha}$, and note that from Eq. (1) we have that $\Delta_t \geq \sqrt{card(x^*)}\|x_t - x^*\|$. As a first step, we are going to show that the point $y_t := x_t + \Delta_t(v_t^+ - v_t^-)$ satisfies: $y_t \cdot \nabla f(x_t) \leq x^* \cdot \nabla f(x_t)$. From Lemma 2 it follows that we can write $x$ as a convex combination $x_t = \sum_{i=1}^{k} \lambda_i v_i$ and write $x^*$ as $x^* = \sum_{i=1}^{k}(\lambda_i - \Delta_i)v_i + \sum_{i=1}^{k} \Delta_i z$, where $\Delta_i \in [0, \lambda_i]$, $z \in \mathcal{P}$, and $\sum_{i=1}^{k} \Delta_i \leq \Delta_t$. It holds that

$$(y_t - x_t) \cdot \nabla f(x_t) = \Delta_t(v_t^+ - v_t^-) \cdot \nabla f(x_t) \leq \sum_{i=1}^{k} \Delta_i(v_t^+ - v_t^-) \cdot \nabla f(x_t)$$

$$\leq \sum_{i=1}^{k} \Delta_i(z - v_i) \cdot \nabla f(x_t) = (x^* - x_t) \cdot \nabla f(x_t),$$

where the first inequality follows since $(v_t^+ - v_t^-) \cdot \nabla f(x_t) \leq 0$, and the second inequality follows from the optimality of $v_t^+$ and $v_t^-$ (Observation 1). Rearranging, we have that indeed

$$\left(x_t + \Delta_t(v_t^+ - v_t^-)\right) \cdot \nabla f(x_t) \leq x^* \cdot \nabla f(x_t). \tag{2}$$

Observe now that from the definition of $\tilde{\eta}_t$ it follows for any $t \geq 1$ that $\frac{\eta_t}{2} \leq \tilde{\eta}_t \leq \eta_t$. Using the smoothness of $f(x)$ we have that

$$
\begin{aligned}
h_{t+1} &= f(x_t + \tilde{\eta}_t(v_t^+ - v_t^-)) - f(x^*) \leq h_t + \tilde{\eta}_t(v_t^+ - v_t^-) \cdot \nabla f(x_t) + \frac{\tilde{\eta}_t^2 \beta}{2}\|v_t^+ - v_t^-\|^2 \\
&\leq h_t + \tilde{\eta}_t(v_t^+ - v_t^-) \cdot \nabla f(x_t) + \frac{\tilde{\eta}_t^2 \beta D^2}{2} \leq h_t + \frac{\eta_t}{2}(v_t^+ - v_t^-) \cdot \nabla f(x_t) + \frac{\eta_t^2 \beta D^2}{2} \\
&= h_t + \frac{\eta_t}{2\Delta_t}\left((x_t + \Delta_t(v_t^+ - v_t^-) - x_t) \cdot \nabla f(x_t) + \frac{\eta_t^2 \beta D^2}{2}\right. \\
&\leq h_t + \frac{\eta_t}{2\Delta_t}(x^* - x_t) \cdot \nabla f(x_t) + \frac{\eta_t^2 \beta D^2}{2} \leq h_t - \frac{\eta_t}{2\Delta_t}h_t + \frac{\eta_t^2 \beta D^2}{2},
\end{aligned}
$$

where the third inequality follows since $(v_t^+ - v_t^-) \cdot \nabla f(x_t) \leq 0$, the forth inequality follows from Eq. (2), and the last inequality follows from convexity of $f(x)$. Finally, plugging the value of $\Delta_t$ completes the proof. $\qquad\square$

## 5 Experiments

In this section we illustrate the performance of our algorithm in numerical experiments. We use the two experimental settings from [16], which include a constrained Lasso problem and a video co-localization problem. In addition, we test our algorithm on a learning problem related to an optical character recognition (OCR) task from [23]. In each setting we compare the performance of our algorithm (DICG) to standard conditional gradient (CG), as well as to the fast away (ACG) and pairwise (PCG) variants [16]. For the baselines in the first two settings we use the publicly available code from [16], to which we add our own implementation of Algorithm 3. Similarly, for the OCR problem we extend code from [20], kindly provided by the authors. For all algorithms we use line-search to set the step size.

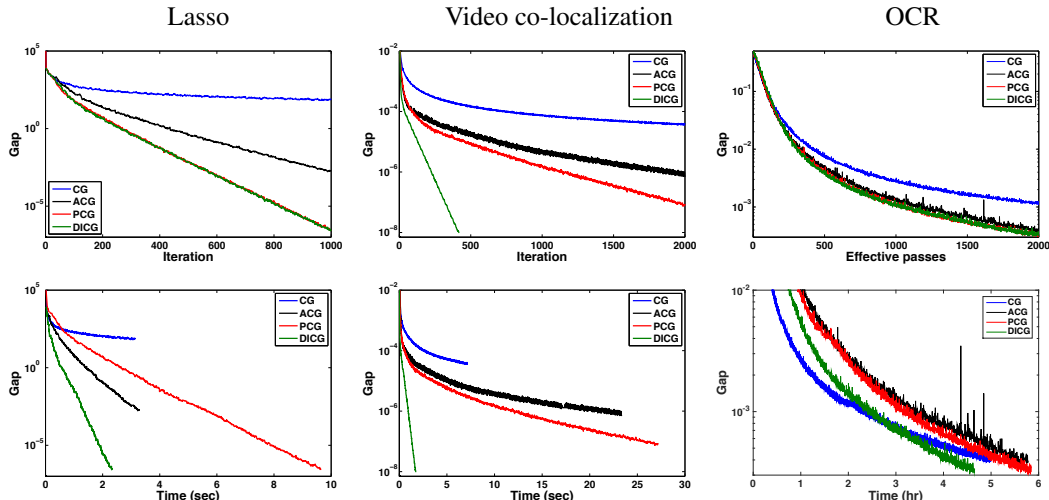

Figure 1: Duality gap $g_t$ vs. iterations (top) and time (bottom) in various settings.

**Lasso**  In the first example the goal is to solve the problem: $\min_{x \in \mathcal{M}} \|\bar{A}x - \bar{b}\|^2$, where $\mathcal{M}$ is a scaled $\ell_1$ ball. Notice that the constraints $\mathcal{M}$ do not match the required structure of $\mathcal{P}$, however, with a simple change of variables we can obtain an equivalent optimization problem over the simplex. We generate the random matrix $\bar{A}$ and vector $\bar{b}$ as in [16]. In Figure 1 (left, top) we observe that our algorithm (DICG) converges similarly to the pairwise variant PCG and faster than the other baselines. This is expected since the away direction $v^-$ in DICG (Algorithm 3) is equivalent to the away direction in PCG (Algorithm 2) in the case of simplex constraints.

**Video co-localization**  The second example is a quadratic program over the flow polytope, originally proposed in [15]. This is an instance of $\mathcal{P}$ that is mentioned in Section A in the appendix. As can be seen in Figure 1 (middle, top), in this setting our proposed algorithm significantly outperforms the baselines, as a result of finding a better away direction $v^-$. Figure 1 (middle, bottom) shows convergence on a time scale, where the difference between the algorithms is even larger. One reason for this difference is the costly search over the history of vertices maintained by the baseline algorithms. Specifically, the number of stored vertices grows fast with the number of iterations and reaches 1222 for away steps and 1438 for pairwise steps (out of 2000 iterations).

**OCR**  We next conduct experiments on a structured SVM learning problem resulting from an OCR task. The constraints in this setting are the marginal polytope corresponding to a chain graph over the letters of a word (see [23]), and the objective function is quadratic. Notice that the marginal polytope has a concise characterization in this case and also satisfies our assumptions (see Section A in the appendix for more details). For this problem we actually run Algorithm 3 in a block-coordinate fashion, where blocks correspond to training examples in the dual SVM formulation [17, 20]. In Figure 1 (right, top) we see that our DICG algorithm is comparable to the PCG algorithm and faster than the other baselines on the iteration scale. Figure 1 (right, bottom) demonstrates that in terms of actual running time we get a noticeable speedup compared to all baselines. We point out that for this OCR problem, both ACG and PCG each require about 5GB of memory to store the explicit decomposition in the implementation of [20]. In comparison, our algorithm requires 220MB of memory to store the current iterate, and the other variables in the code require 430MB (common to all algorithms), so using DICG results in significant memory savings.

# 6  Extensions

Our results are readily extendable in two important directions. First, we can relax the strong convexity requirement of $f(x)$ and handle a broader class of functions, namely the class considered in [2]. Second, we extend the line-search variant of Algorithm 3 to handle arbitrary polytopes, but without convergence guarantees, which is left as future work. Both extensions are brought in full detail in Section C in the appendix.

## Footnotes

[1]While the convergence rate of this pairwise variant, established in [16], is significantly worse than other away-step-based variants, here we show that a proper analysis yields state-of-the-art performance guarantees.

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
