[Supplementary Material]

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

# A  Examples of Polytopes

In this section we turn to survey several important examples of structured polytopes that fit the assumptions in Subsection 2.1 and detail the application of Algorithm 3 to optimization over these polytopes.

**Unit Simplex**    The simplex in $\mathbb{R}^n$ is the set of all distributions over $n$ elements, i.e. the set:

$$\mathcal{S}_n = \{x \in \mathbb{R}^n \,|\, x \geq 0\,, \sum_{i=1}^n x_i = 1\}.$$

Alternatively, $\mathcal{S}_n$ is the convex hull of all standard basis vectors in $\mathbb{R}^n$.

It is easy to verify that $D = \sqrt{2}$.

Linear minimization over the simplex is trivial and can be carried out by a single pass over the non-zero elements in the linear objective. In particular, computing $v_t^-$ in Algorithm 3 simply amounts to finding the largest (signed) entry in $\nabla f(x_t)$ which corresponds to a non-zero entry in $x_t$, and thus is even more efficient than computing the standard CG direction $v_t^+$.

**Flow polytope**    Let $G$ be a *directed acyclic graph* (DAG) with a set of vertices $V$ such that $|V| = n$, and a set of edges $E$ such that $|E| = m$, and let $s, t$ be two vertices in $V$ which we refer to as the *source* and the *target*, respectively. The $s - t$ flow polytope, denoted here by $\mathcal{F}_{st}$, is the set of all unit $s - t$ flows in $G$, where for each point $x \in \mathcal{F}_{st}$ and $i \in [m]$, the entry $x_i$ is the amount of flow through edge $i$ according to the flow $x$. $\mathcal{F}_{st}$ is also known as the $s - t$ *path polytope* since it is the convex hull of all identifying vectors of paths from $s$ to $t$ in the graph $G$. It is easy to verify that that $D < \sqrt{2n}$.

Since $\mathcal{F}_{st}$ is the convex hull of paths, linear minimization is straightforward: given a linear objective $c \in \mathbb{R}^m$, we need to find the identifying vector of the lightest $s - t$ path in $G$ with respect to the edge weights induced by $c$. Since the graph $G$ is a DAG, this could be carried out in $O(m)$ time [21]. In particular, computing the direction $v_t^-$ in Algorithm 3 over the flow polytope, amounts to finding the lightest $s - t$ path in $G$ with respect to the gradient vector $\nabla f(x_t)$, under the constraint that all edges on the path are assigned non-zero flow by $x_t$. Thus, we can compute $v_t^-$ by running a shortest $s - t$ path algorithm after removing all edges with zero flow from the graph. Thus, as in the simplex case, computing $v_t^-$ is even more efficient than computing the standard direction $v_t^+$.

It is also important to note that when $G$ is not extremely sparse, i.e., when $m = \omega(n)$, it holds for every vertex $v$ of $\mathcal{F}_{st}$ that $\text{card}(v) << m$. Thus, if $x^*$ can be expressed as a combination of only a few paths, i.e., it corresponds to a sparse flow, it holds that $\text{card}(x^*)$ is much smaller than the standard dimension of the problem $m$.

**Perfect Matchings polytope**    Let $G$ be a *bipartite* graph with $n$ vertices on each side and $m$ crossing edges. The perfect matching polytope, denoted here by $\mathcal{M}$, is the convex hull of all identifying vectors of perfect matchings in $G$. In case the two sides of $G$ are fully connected, this polytope is also known as the Birkhoff polytope - the set of all $n \times n$ doubly stochastic matrices, i.e. matrices with non-negative real entries whose entries along any row and any column add up to 1. It easily follows that $D \leq \sqrt{2n}$.

In order to minimize a linear objective over $\mathcal{M}$, we need to find a minimum-weight perfect matching in a bipartite graph, where the edge weights are induced by the linear objective. This could be carried out via combinatorial algorithms in $\min\{\tilde{O}(\sqrt{n}m), O(n^3)\}$ time [21]. As in the flow polyope, in this case also, computing $v_t^-$ is even more efficient than computing $v_t^+$, since it amounts to finding a minimum weight perfect matching after all edges that are zero-valued in $x_t$ are removed from the graph.

As in the flow polytope, in case $G$ is not trivially sparse, i.e., when $m = \omega(n)$, it holds that if $x^*$ could be expressed as a combination of only a few matchings in $G$, then $\text{card}(x^*) << m$, where $m$ is the dimension of the problem.

**Marginal polytope**    In *Graphical Models* several optimization problems are defined for variables representing marginal distributions over subsets of model variables. There exists a set of linear

constraints, known as the *marginal polytope*, which guarantees that these variables are legal marginals of some global distribution [24]. For example, the learning problem in Max-Margin Markov Networks is defined as a quadratic program over the marginal polytope [23].

For general graphical models the marginal polytope consists of an exponential number of constraints. Fortunately, for some models, such as tree-structured graphs, the polytope can be characterized by a polynomial number of local consistency constraints, known as the *local marginal polytope* [24]. Consider a set of discrete variables $(y_1, \ldots, y_n)$, and denote by $\mu_c(y_c)$ the marginal probability of an assignment to a subset of these variables $y_c$. Then the local marginal polytope is defined as:

$$\mathcal{M}_L = \left\{ \mu \geq 0 : \begin{array}{ll} \sum_{y_{c \setminus i}} \mu_c(y_c) = \mu_i(y_i) & \forall c, i \in c, y_i \\ \sum_{y_i} \mu_i(y_i) = 1 & \forall i \end{array} \right\}$$

For tree-structured graphs $\mathcal{M}_L$ is known to have only integral vertices [24], so it has the desired form assumed in Section 2.1.

In this case $D = \sqrt{2}|\mathcal{C}|$, where $\mathcal{C}$ is the number of subsets $y_c$ (factors in the graphical model).

In many interesting cases linear optimization over the marginal polytope can be implemented efficiently via dynamic programming. For example, for chain-structured graphs the Viterbi algorithm is used. Finally, we note that computing the direction $v_t^-$ in Algorithm 3 can often be cheaper than computing $v_t^+$, since the restriction to the support of $x_t$ can eliminate many of the possible configurations of marginals.

# B    Detailed Proofs from Section 4

## B.1    Proof of Observation 2

*Proof.* From the optimality of $v_t^-$ it follows that for any $i \in [n]$, if $x_t(i) = 0$, then $v_t^-(i) = 0$ (note in particular that any vertex with positive weight in some convex decomposition of $x_t$ must satisfy this condition). Thus, from our assumption on the size of positive entries in $x_t$, and since $v_t^- \in \{0, 1\}^n$, it follows that the vector $w_t := x_t - \tilde{\eta}_t v_t^-$, satisfies: $w_t \geq 0$. Since $v_t^+$ is feasible it also follows that $x_{t+1} = w_t + \tilde{\eta}_t \geq 0$. Finally, since $x_t, v_t^-, v_t^+$ are all feasible, it also holds that $Ax_{t+1} = b$. Thus, $x_{t+1}$ is feasible. □

## B.2    Proof of Lemma 2

*Proof.* Consider writing $y$ as a convex combination of vertices, $y = \sum_{i=1}^s \gamma_i u_i$ for some appropriate integer $s$. Applying Lemma 5.3. from [8], it follows that we can write $x$ as

$$x = \sum_{i=1}^s (\gamma_i - \tilde{\Delta}_i)u_i + (\sum_{i=1}^s \tilde{\Delta}_i)\tilde{z}, \tag{3}$$

where $\tilde{\Delta}_i \in [0, \gamma_i] \, \forall i \in [s]$, $\tilde{z} \in \mathcal{P}$, and for every $i$ with $\tilde{\Delta}_i > 0$ there exists $j_i \in [n]$ such that $\tilde{z}(j_i) = 0$ and $u_i(j_i) > 0$. Since each $u_i$ is a vertex of $\mathcal{P}$ and thus a point of the $\{0, 1\}$-hypercube, it further follows that $u_i(j_i) = 1$. Let $C = \{j_i \, | \, i \in [s]\}$. Observe that $|C| \leq \text{card}(y)$. Now, we have that

$$\|x - y\|^2 = \|\sum_{i=1}^s \tilde{\Delta}_i(u_i - \tilde{z})\|^2 \geq \sum_{j \in C} \left( \sum_{i=1}^s \tilde{\Delta}_i(u_i(j) - \tilde{z}(j)) \right)^2 = \sum_{j \in C} \left( \sum_{i=1}^s \tilde{\Delta}_i u_i(j) \right)^2$$

$$\geq \frac{1}{|C|} \left( \sum_{j \in C} \sum_{i=1}^s \tilde{\Delta}_i u_i(j) \right)^2 \geq \frac{1}{|C|} \left( \sum_{i=1}^s \tilde{\Delta}_i \right)^2.$$

Rearranging we have that $\sum_{i=1}^s \tilde{\Delta}_i \leq \sqrt{|C|}\|x - y\| \leq \sqrt{\text{card}(y)}\|x - y\|$.

Note that using the convex decomposition of $x$ as in Eq. (3), and the above bound, it follows that we can rewrite $y$ as a convex decomposition as suggested in the lemma. □

## B.3 Proof of Theorem 1

*Proof.* We are first going to prove the convergence rate stated in the theorem, assuming that all iterates are feasible. Then we will show that for our choice of step-sizes, indeed the iterates are feasible. We will prove by induction that there exist $c_0, c_1$ such that for all $t \geq 1$ it holds that $h_t \leq c_0(1 - c_1)^{t-1}$. Clearly for the base case we must require that $c_0 \geq h_1$.

Suppose now that the induction holds for some $t \geq 1$. Let us set

$$\eta_t = \frac{M_1}{2M_2}\sqrt{c_0}(1 - c_1)^{\frac{t-1}{2}}. \tag{4}$$

Using Lemma 3 and the induction hypothesis we have that

$$
\begin{aligned}
h_{t+1} &\leq h_t - \frac{M_1^2}{2M_2}\sqrt{c_0(1 - c_1)^{t-1}}h_t^{1/2} + \frac{M_1^2}{4M_2}c_0(1 - c_1)^{t-1} \\
&\leq h_t - \frac{M_1^2}{2M_2}h_t + \frac{M_1^2}{4M_2}c_0(1 - c_1)^{t-1} \\
&= h_t\left(1 - \frac{M_1^2}{2M_2}\right) + \frac{M_1^2}{4M_2}c_0(1 - c_1)^{t-1} \\
&\leq c_0(1 - c_1)^{t-1}\left(1 - \frac{M_1^2}{4M_2}\right),
\end{aligned}
$$

where the induction hypothesis was used in both the second and third inequalities. In the third inequality we have also used the fact that

$$\frac{M_1^2}{2M_2} = \frac{\alpha}{8\beta\mathrm{card}(x^*)D^2} < 1, \tag{5}$$

where the inequality follows since $\alpha \leq \beta$ and both $\mathrm{card}(x^*), D$ are at least 1.

Thus, if we set $c_1 = \frac{M_1^2}{4M_2}$, the induction follows.

We now turn to figure out $c_0$.

Using the smoothness of $f(x)$ and the choice of $x_1$ in Algorithm 3, we have that

$$
\begin{aligned}
h_1 &= f(x_1) - f(x^*) = f(x_0 + (x_1 - x_0)) - f(x^*) \\
&\leq f(x_0) - f(x^*) + (x_1 - x_0) \cdot \nabla f(x_0) + \frac{\beta\|x_0 - x_1\|^2}{2} \\
&\leq f(x_0) - f(x^*) + (x^* - x_0) \cdot \nabla f(x_0) + \frac{\beta\|x_0 - x_1\|^2}{2} \leq \frac{\beta D^2}{2},
\end{aligned}
$$

where the last inequality follows from the convexity of $f(x)$.

Thus, we can set $c_0 = \frac{\beta D^2}{2} = M_2$, which completes the proof of the convergence rate.

Now, it remains to prove that indeed all iterates are feasible. First note that the sequence $\{\eta_t\}_{t \geq 1}$, as defined in Eq. (4) is monotonically non-increasing. Furthermore, plugging the values $M_1, M_2, c_0$, we have that

$$\eta_1 = \frac{M_1\sqrt{c_0}}{2M_2} = \frac{1}{2}\sqrt{\frac{M_1^2}{M_2}} = \frac{1}{2}\sqrt{\frac{\alpha}{4\beta D^2\mathrm{card}(x^*)}} \leq 1,$$

where the inequality follows similarly to the one in Eq. (5). Thus, our choice of step-size sequence $\{\eta_t\}_{t \geq 1}$ satisfies the conditions of Lemma 1, and thus it follows that all iterates of Algorithm 3 are feasible. $\qquad \square$

## B.4 Linear convergence of duality gap

The following corollary of Theorem 1 shows that the so-called *duality gap*, which serves as a certificate for the sub-optimality of the iterates of Algorithm 3, also converge with a linear rate.

**Corollary 1.** *For any iteration $t$ of Algorithm 3, define the dual gap $g_t := (x_t - v_t^+) \cdot \nabla f(x_t)$, and observe that, since $f(x)$ is convex, $h_t \le g_t$. Then, for any $t$ which satisfies: $h_t \le \frac{\beta D^2}{2}$, it holds that*

$$g_t \le \sqrt{2\beta D^2 h_t}.$$

*Proof.* Fix an iteration $t$. Using the $\beta$-smoothness of $f(x)$ we have that

$$\forall \eta \in (0, 1]: \quad f(x^*) \le f(x_t + \eta(v_t^+ - x_t)) \le f(x_t) + \eta(v_t^+ - x_t) \cdot \nabla f(x_t) + \frac{\eta^2 \beta D^2}{2}.$$

Rearranging, we have that

$$\forall \eta \in (0, 1]: \quad g_t = (x_t - v_t^+) \cdot \nabla f(x_t) \le \frac{1}{\eta} h_t + \frac{\eta \beta D^2}{2}.$$

Thus, when $\sqrt{\frac{2h_t}{\beta D^2}} \le 1$, we can set $\eta = \sqrt{\frac{2h_t}{\beta D^2}}$ in the above inequality, and obtain the corollary. $\quad\square$

## C   Extensions

In this section we detail two extensions of our result: i) relaxing the specific structure of the polytope $\mathcal{P}$ considered in Subsection 2.1, and ii) relaxing the strong convexity requirement on the objective function $f(x)$.

### C.1   Extension of Algorithm 3 to arbitrary polytopes

In this subsection we detail how to extend our approach to a broader class of polytopes. While proving rigorous guarantees for this extension is beyond the scope of this paper and left for future work, the encouraging experimental results for Algorithm 3 with line-search, suggest that this extended variant, for which line-search is also possible, may also exhibit favorable empirical performance. Towards this end, in this subsection we consider minimizing a smooth and strongly-convex function over an arbitrary polytope $\mathcal{P}$ which we assume is given in the following way:

$$\mathcal{P} = \{x \in \mathbb{R}^n \mid A_1 x = b_1, \, A_2 x \le b_2\},$$

where $A_2$ is $m \times n$. We assume that given a point $x \in \mathbb{R}^n$, we have an efficient way to evaluate the vector $A_2 x$, which is indeed the case for most structured polytopes of interest.

---

**Algorithm 4** Decomposition-invariant Pairwise Conditional Gradient with Line-search for Arbitrary Polytopes

---

1: let $x_0$ be an arbitrary point in $\mathcal{P}$
2: $x_1 \leftarrow \arg\min_{v \in \mathcal{V}} v \cdot \nabla f(x_0)$
3: **for** $t = 1...$ **do**
4: $\quad v_t^+ \leftarrow \arg\min_{v \in \mathcal{V}} v \cdot \nabla f(x_t)$
5: $\quad$ define the vector $c \in \mathbb{R}^m$ as follows:

$$c_i := \begin{cases} 0 & \text{if } A_2(i) \cdot x_t < b_2(i) \\ \infty & \text{if } A_2(i) \cdot x_t = b_2(i) \end{cases}$$

6: $\quad v_t^- \leftarrow \arg\min_{v \in \mathcal{V}} (-\nabla f(x_t)) \cdot v + c^\top A_2 v$
7: $\quad \gamma_t \leftarrow \max\{\gamma \in [0, 1] \mid A_2(x_t + \gamma(v_t^+ - v_t^-)) \le b_2\}$
8: $\quad \eta_t \leftarrow \arg\min_{\eta \in [0, \gamma_t]} f(x_t + \eta(v_t^+ - v_t^-))$
9: $\quad x_{t+1} \leftarrow x_t + \eta_t(v_t^+ - v_t^-)$
10: **end for**

---

**Observation 3** (optimal away-step for an arbitrary polytope). *Consider an iteration $t$ of Algorithm 4 and suppose that the iterate $x_t$ is feasible. Let $x_t = \sum_{i=1}^k \lambda_i v_i$ for some integer $k$, be an irreducible way of writing $x_t$ as a convex sum of vertices of $\mathcal{P}$, i.e., $\lambda_i > 0$ for all $i \in [k]$. Then it holds that*

$$\forall i \in [k]: \quad v_i \cdot \nabla f(x_t) \le v_t^- \cdot \nabla f(x_t), \qquad \gamma_t > 0.$$

*Moreover, there exists a convex decomposition of $x_t$ that assigns a weight at least $\gamma_t$ to $v_t^-$.*

*Proof.* Let $x_t = \sum_{i=1}^{k} \lambda_i v_i$ be a decomposition of $x_t$ into vertices of $\mathcal{P}$ such that $\lambda_i > 0$ for all $i \in [k]$. Observe that for any $j \in [m]$ and $i \in [k]$ it holds that $A_2(j) \cdot x_t = b_2(j) \Rightarrow A_2(j) \cdot v_i = b_2(j)$. Note that by definition of the vector $c$ and $v_t^-$ it holds that

$$
\begin{aligned}
v_t^- &\in \arg\max_{v \in \mathcal{V}} \nabla f(x_t) \cdot v - c^\top A_2 v \equiv \arg\max_{v \in \mathcal{V}} \nabla f(x_t) \cdot v + c^\top (b_2 - A_2 v) \\
&\equiv \arg\max_{v \in \{y \in \mathcal{V} \mid \forall j \in [m]: A_2(j) \cdot x_t = b_2(j) \Rightarrow A_2(j) \cdot y = b_2(j)\}} v \cdot \nabla f(x_t).
\end{aligned}
\tag{6}
$$

Thus, it follows that for all $i \in [k]$, $v_t^- \cdot \nabla f(x_t) \geq v_i \cdot \nabla f(x_t)$.

In order to prove the second part of the observation, we note that from the RHS of Eq. (6) it follows that there exists $\gamma_t > 0$ such that indeed $x_t - \gamma_t v_t^- \leq (1 - \gamma_t) b_2$. To see this, consider some $j \in [m]$. If $A_2(j) \cdot x_t = b_2(j)$, then from the RHS of Eq. (6), it follows that $A_2(j) \cdot v_t^- = b_2(j)$ and thus, for any $\gamma_t$ it holds that $A_2(j) \cdot (x_t - \gamma_t v_t^-) = (1 - \gamma_t) b_2(j)$. Otherwise, there exists some $\epsilon_j > 0$ such that $A_2(j) \cdot v_t^- \leq b_2(j) - \epsilon_j$. Thus, for small enough, yet positive $\gamma_t$ we will have that $A_2(j) \cdot (x_t - \gamma_t v_t^-) \leq (1 - \gamma_t) b_2(j)$. Since it clearly also holds that $A_1(x_t - \gamma_t v_t^-) = (1 - \gamma_t) b_1$, we have that the vector $w_t := x_t - \gamma_t v_t^-$ satisfies: $w_t \in (1 - \gamma_t) \mathcal{P}$. Hence, $w_t$ can be decomposed as $w_t = \sum_{i=1}^{q} \tilde{\gamma}_i \tilde{v}_i$, where $q$ is a positive integer and for all $i \in [q]$, $\tilde{\lambda}_i > 0$, $\tilde{v}_i$ is a vertex of $\mathcal{P}$, and $\sum_{i=1}^{q} \tilde{\lambda}_i = 1 - \gamma_t$. Thus, since $v_t^-$ is a vertex of $\mathcal{P}$, it follows that $x_t = w_t + \gamma_t v_t^-$ admits the convex decomposition $\sum_{i=1}^{q} \tilde{\lambda}_i \tilde{v}_i + \gamma_t v_t^-$, as needed. $\square$

The following lemma is an immediate consequence of the choice of $\gamma_t$ in Algorithm 4.

**Lemma 4.** *The iterates of Algorithm 4 are always feasible.*

## C.2 Relaxing the strong convexity of the objective function

Until now we have assumed that the objective function $f$ is strongly convex. However, as can be observed from our analysis, the only consequence of strong convexity that we relied on in our analysis, is Eq. (1). Indeed, there exist functions which are not strongly convex, that under certain conditions, still satisfy Eq. (1), and thus are compatible with our method and analysis.

Following the work of Beck and Shtern [2], we can consider a broader class of objective functions, namely functions that take the following form:

$$
f(x) = g(\hat{A}x) + \hat{b} \cdot x,
\tag{7}
$$

where $\hat{A} \in \mathbb{R}^{m \times n}$, and $g : \mathbb{R}^m \to \mathbb{R}$ is smooth and strongly convex.

In [2] (Lemma 2.5) it was shown, using an application of Hoffman's lemma, that there exists a constant $\kappa$ which depends both on the condition number of $g$ and the parameters $\hat{A}, \hat{b}$, such that for any feasible point $x$, it holds that

$$
\min_{y \in \mathcal{P}^*} \|x - y\|^2 \leq \kappa \left( f(x) - f^* \right),
\tag{8}
$$

where $\mathcal{P}^* \subset \mathcal{P}$, is the set of all feasible points that minimize $f(x)$ over $\mathcal{P}$, and $f^*$ is the minimum value of $f(x)$ over $\mathcal{P}$.

It is easy to verify that Eq. (8) can be readily used in our analysis instead of Eq. (1), and thus our results extend to handle objectives of the form given in Eq. (7).

We note that now, the dependency in our analysis and in Theorem 1 on the strong convexity parameter $\alpha$ will be replaced with $\kappa$, and the dependency on $\text{card}(x^*)$ will be replaced with $\max_{y \in \mathcal{P}^*} \text{card}(y)$.

## D Lower Bound for Problems with a Sparse Solution

In this section we present a simple lower bound on the approximation error of, informally speaking, any natural conditional gradient variant that when initialized with a vertex of the feasible set, its iterate after $t$ iterations admits a convex decomposition into at most $t + 1$ vertices of the polytope. That is, on each iteration at most a single new vertex is added to the decomposition. The lower bound shows that there exists a 1-smooth and 1-strongly convex function $f$, for which, any such CG variant

which is applied to the minimization of $f$ over the unit simplex, must take $\Omega(\text{card}(x^*))$ steps before entering the linear convergence regime. To date, none of the previous analyses of linearly converging CG variants matches this lower bound since, in this exact setting, they all require, in worst-case, $\Omega(n)$ steps before entering the linear convergence regime, i.e., number of steps that is independent of $\text{card}(x^*)$.

To the best of our knowledge, Algorithm 3 and the corresponding Theorem 1 are the first to match this lower bound. We emphasize that the idea behind the construction of this lower bound is well known and follows almost immediately from previous constructions, such as those in [13, 8].

**Lemma 5.** *Fix an even integer $k \in [n]$, and consider the optimization problem*

$$\min_{x \in \mathcal{S}_n} \{ f(x) := \frac{1}{2} \| x - \frac{1}{k} \boldsymbol{1}_k \|^2 \},$$

*where $\mathcal{S}_n$ denotes the unit simplex in $\mathbb{R}^n$, i.e., $\mathcal{S}_n := \{ x \in \mathbb{R}^n \mid x \geq 0, \|x\|_1 = 1 \}$, and $\boldsymbol{1}_k$ is a vector in $\mathbb{R}^n$, defined as:*

$$\boldsymbol{1}_k(i) = \begin{cases} 1 & \text{if } 1 \leq i \leq k \\ 0 & \text{else} \end{cases}$$

*Observe that $x^* = \frac{1}{k} \boldsymbol{1}_k$ is the unique minimizer of $f$ over $\mathcal{S}_n$. Then, any point $x \in \mathcal{S}_n$, for which it holds that $\text{card}(x) \leq k/2$ satisfies:*

$$f(x) - f(x^*) \geq \frac{1}{4k}.$$

*Proof.* Fix a point $x \in \mathcal{S}_n$ for which it holds that $\text{card}(x) \leq k/2$. In order to lower bound the approximation error of $x$, it suffices to consider the entries which are zero for $x$ and non-zero for $x^*$. Thus, we have that

$$f(x) \geq \frac{1}{2} \cdot \frac{k}{2} \cdot \left( 0 - \frac{1}{k} \right)^2 = \frac{1}{4k}.$$

$\square$