[Reviews · NeurIPS 2016]

Reviewer 1

Summary

The paper proposes a new variant of Frank-Wolfe which gets rid of the memory aspect in the decomposition-based variants. Furthermore, an improved rate is presented for the case of a sparse optimal solution. The used proof techniques are novel and useful.

Qualitative Assessment

The paper proposes a new variant of Frank-Wolfe which gets rid of the memory aspect in the decomposition-based variants. Furthermore, an improved rate is presented for the case of a sparse optimal solution. The used proof techniques are novel and useful. A very nice contribution of the paper is that it significantly deepens the insights into pairwise FW variants, such as getting rid of the |V|! artifact present in [16]. The proof techniques are novel and might be useful for the future, in particular since the algorithm variants considered in the paper here are simple and natural. The presented preliminary experimental results are convincing for the presented new algorithm. A concern is the lost translation invariance of the method, as this card(x^*). Nevertheless I think the more explicit dependence on card(x^*) is valuable and gives new insights. The paper could be strengthened even further by also discussing different possible representations of the input problem, i.e. how do all convergence constants change when e.g. parameterizing the problem in terms of the convex combination of vertices (n= |V|) vs when using an explicit embedding dimension n holding all V and iterates. Another slight concern I have is that in many places the authors state a quadratic complexity in n for competing methods, such as in Table 1 and around line 80. "Maintaining such a decomposition and computing the away-steps require both memory and per-iteration runtime overheads that are at least quadratic in the dimension." This is not precise as written now, as the cost in a naive implementation is the number of iterations t times the dimension n, where the former one is typically way smaller than n. Furthermore, the pairwise variant of the FW algorithm should already be mentioned in this context, again only having a weak dependence on t in this respect. In other word, in practice, users of such algorithms will never use Carathéodory’s theorem to update a decomposition. Instead, one simply maintains both the decomposition and the iterate (e.g. in Algorithm 2), and performs a constant number of updates of the scalar weights of the decomposition. That is two of them in the pairwise case, and a scalar and the new weight in case of away steps). The discussion around lines 158 should better reflect both t < n and t > n regimes, or be tuned down accordingly. (Also, definition of n should appear before it is used in Table 1). In this same line of thought, sure I see that the presented algorithm gives the novelty of decomposition-invariance, but I'm not fully convinced yet by the presented arguments why this property should be so desirable in practice. I agree that the property is nice as mentioned around lines 166, but the benefit is not quantified at this stage. In the experiments, it needs to be explained if the competing methods are implemented using a Carathéodory type update or the simpler explicit decomposition. ** Minor: Line 150 'bring'? Algorithm 3 'smallest natural..' Algorithm 3: case distinction xt > 0 misses the dependence on i! UPDATE AFTER AUTHOR FEEDBACK: Thanks, keeping my assessment for now. The min(T,n) change is appreciated.

Confidence in this Review

3-Expert (read the paper in detail, know the area, quite certain of my opinion)


Reviewer 2

Summary

The authors present a new variant of away-step Frank Wolfe algorithms for a special class of polytope (whose vertices lie on hypercube), allowing to get rid of the requirement on vertex decomposition and rate dependency on dimension compared to existing linearly convergent FW algorithms. Convergence results are well established with explicit comparison to existing work; diverse experiments are provided showing convincing evidence of outperformance.

Qualitative Assessment

I found the paper a rather pleasure to read; it comes with clear motivation and intuition that guides the reader through. The new algorithm breaks the two bottlenecks of usual away-step FW algorithms. Although it only applies to a restricted class of polytopes, but the results are interestingly new, and could be of great practical importance. My only minor comments are for the experiment part. I would suggest the authors to move the discussion on generalizing to non-strongly convex before the experiment, since some instances, e.g. Lasso, do not necessarily meet the strongly convexity assumed in the main text. Also, details of the experimental setups, e.g. size of problem, magnitude of the intrinsic parameters like sparsity, should be provided at least in the supplementary to better reflect the speed shown in the figures. == post-rebuttal answer== I have read the author's rebuttal.

Confidence in this Review

2-Confident (read it all; understood it all reasonably well)


Reviewer 3

Summary

This paper gives a faster Frank-Wolfe type algorithm for minimizing strongly convex smooth function over {0,1} polytope. All previous Frank-Wolfe type algorithms that has linear convergence takes quadratic time per iteration. However, this algorithm takes only linear time per iteration. Furthermore, this algorithm converges faster if the solution is sparse and converges the same otherwise.

Qualitative Assessment

Overall this paper is very good in both theory and experiment. The only reason I didn't give an even higher score is due to the fact, in many applications, the bound is still like |x*|_0 n kappa log(1/eps) which is still quite large compare to the lower bound n log(1/eps) in the oracle model.

Confidence in this Review

2-Confident (read it all; understood it all reasonably well)


Reviewer 4

Summary

A variant of conditional gradient aka Frank Wolfe algorithm have been proposed. The authors showed linear convergence of the proposed algorithm for structured polytopes like Simplex, Flow, Perfect Matchings and Marginal polytopes. Additionally, this method also uses less memory compared to others. Like most CG methods, two different step size policies are suggested with one of them being "deterministic" while the other is based off on line search.

Qualitative Assessment

The paper is well written in general, but I think that the introduction can be made shrunk a bit. Table 1 makes it easier to compare different algorithms both time and memory complexity. The results are nice and a good direction to pursue in terms of conditional gradient algorithms, the paper as of its current form needs more work as the convergence analysis is valid is only valid for the class of polytopes mentioned in the paper. While the time complexity for arbitrary polytopes might not be as good as stated in table 1, some weaker convergence (more than what is mentioned in supplement) would tremendously benefit the applicability of the algorithm proposed if one chooses to use conditional gradient type algorithms with the modifications proposed. The algorithm is well presented and the proofs are easy to follow. Some lemmas are presented in the main paper whereas others are in the supplement. Since the proofs do not reveal much about the algorithms, a sketch of the proofs will be sufficient in the main paper. For example, the main intuition for lemma 3. is in lines 246- 248. Experiments on OCR problem speaks for the strength of the proposed algorithms but I am not convinced with the experimental results because the class of problems that can be solved is still very small compared to say [1]. Secondly, given that there are good solvers/algorithms practically for the problems considered in the experiments it only makes sense to compare with those rather than just with other conditional gradient algorithms. 1. A Universal Primal-Dual Convex Optimization Framework (NIPS 2015), Alp Yurtsever, Quoc Tran-Dinh, Volkan Cevher. == Post author feedback == Thanks for clarifying my concerns. I still think that moving the proofs to the supplement and bringing strong convexity part, extension to arbitrary polytopes to the main paper would be useful for practitioners.

Confidence in this Review

2-Confident (read it all; understood it all reasonably well)


Reviewer 5

Summary

The paper proposes a new variant of Pairwise Conditional Gradient (PCG) method. Unlike previous PCG method that searches an "away vertex" over a maintained bases set for the current iterate, the new PCG searches the away vertex via a Linear Optimization Oracle (LMO) given by modified gradient, which reduces time and space complexity significantly for problem of domain comprising large number of vertices. The new approach to find away vertex also helps improve iteration complexity compared to some other recent analysis.

Qualitative Assessment

Fast-convergent variants of Conditional Gradient algorithm have drawn much interest in the recent years. Previous approach however requires maintaining a list of support vertices for the iterate, which results in intractable cost for problem with exponentially large number of vertices, where the number of support vertices could increase linearly with the number of iterations. This paper proposes an elegant solution to this---searching away vertex using Linear Optimization Oracle, which not only reduces per-iteration cost for problem of large number of vertices but also reduces iteration complexity from "proportional to the problem dimension" to "proportional to the support size of optimal solution". The review of literature and presentation of analysis are clear, and the proof is correct as far as I can tell. --------------------after author feedback--------------------- After reading the feedback, I am more confident on the evaluation and would like to keep my previous review scores.

Confidence in this Review

3-Expert (read the paper in detail, know the area, quite certain of my opinion)


Reviewer 6

Summary

In this paper, the authors provide a new conditional-gradient-type method for minimization of a smooth strongly convex function over a special class of polytopes. The proposed method is less memory consuming and have cheaper iteration than the state-of-the-art methods. Convergence rate of the method is provided and numerical comparison with existing methods is also presented.

Qualitative Assessment

Initial review. Technical quality: 1. Line 138: Actually, this statement holds for large n. 2. In my opinion, the arguments about the drawbacks of the existing methods in the end of Section 3.1 are not so strong. Existing methods can have superlinear memory and running-time overheads in general case. But in the considered in this paper case of structured polytopes, it may not hold. For example, in the case of the standard simplex, the step 5 in Algorithm 2 costs $O(n)$ operations. The same holds for the memory consumption. It seems that one can criticize the decomposition-specific performance argument in the same way. 3. It is not clear how one can use logarithmic-scale search to estimate $M_1/sqrt{M_2}$. 4. It is not clear why, for the numerical experiments, the authors choose the "line-search" version of their algorithm, despite that it doesn't have a proved convergence theorem. 5. It is not clear, why the authors do not compare their method with the method of [6]. It seems that the latter can be efficient on the considered class of structured polytopes. Novelty/originality: To the best of my knowledge, the method and its analysis are new. Potential impact or usefulness: 1. A more detailed description of "applications in machine learning, signal processing and other domains" would increase the potential impact of the paper. 2. Potential drawback of the method could be that one needs an information about $card(x^*)$ in order to implement method with provable convergence rate. 3. It seems that the line-search option in Algorithm 3 could be quite hard to implement in practice. Clarity and presentation: 1. It was quite hard for me to clarify, what is meant by minimization of a linear form which is given by a vector with infinite-valued components. My first guess was that if you minimize such a linear form given by a vector $(\infty, 1)^T$, it is the same as minimization of a linear form given by a vector $(1, 0)^T$. The reason is that you can neglect 1 in comparison with $\infty$. 2. The text should be proof readed. Here are some examples of misprints. Line 54: "a the convex" should be "as the convex" Line 92: "shaper" should be "sharper" Step 6 of Algorithm 3: $R^m$ should be $R^n$ ===================================== Update after author feedback Technical quality: 1. This comment remains the same. 2. As it is done now, in the end of Section 3.1, it seems a bit unfair to criticize the existing methods and their performance for _general_ polytopes, and, then, propose a method for a _narrow_ class of polytopes. In that sense, the authors provide in their rebuttal a reasonable argument "However for more complex polytopes ..." It should be included in the text. At the same time, the authors write in the rebuttal "This difference in memory requirement is well demonstrated in our OCR experiment, showing significant savings for our method." But, in the text, they report only memory consumption of ACG and PCG, but not the memory consumption of their method. 3. After clarification in the authors' feedback, I still have some doubts concerning the logarithmic-scale search for estimation of $M_1/sqrt{M_2}$. As far as I understood from the text, it will work if we estimate M_1/sqrt{M_2} by some M \in [0.5 M_1/sqrt{M_2}, M_1/sqrt{M_2}]. This interval does not seem to be very wide. 4. In the rebuttal, the arguments on choice of "line-search" version for experiments seem to be reasonable and it should be added to the text. 5. After clarification in the authors' feedback, this point is still not clear for me. It could be fair to compare the proposed provable algorithm with [6], since both of them do not use line search. Novelty/originality: No change in my opinion. Potential impact or usefulness: 1. This comment remains the same. 2. In their rebuttal, the authors write that there is no need to know $card(x^*)$. I agree with their explanation. Nevertheless, as I think, potential drawback of the method could be that one needs a good approximation for M_1/sqrt{M_2} (i.e. some M \in [0.5 M_1/sqrt{M_2}, M_1/sqrt{M_2}]) in order to implement method with provable convergence rate. 3. I think the remark in authors' feedback "In general, one can apply the line-search to the upper-bound given by Definition 2, which is always easy to compute." should be added to the text with some explanations on why minimizing the upper-bound will work. 4. New item. I think, the description of logarithmic-scale search for estimation of $M_1/sqrt{M_2}$ should be added to the text. Clarity and presentation: 1. I think, the authors can easily clarify this point in the text and should do it. 2. This comment remains the same.

Confidence in this Review

2-Confident (read it all; understood it all reasonably well)